Accepted at the ICLR 2024 Workshop on AI4Differential Equations In Science

# Joint Parameter and Parameterization Inference with Uncertainty Quantification through Differentiable Programming

**Yongquan Qu, Mohamed Aziz Bhouri & Pierre Gentine**
NSF Center for Learning the Earth With Artificial Intelligence and Physics
Department of Earth and Environmental Engineering
Columbia University
New York, NY 10027, USA
`{yq2340,mb4957,pg2328}@columbia.edu`

## Abstract

Accurate representations of unknown and sub-grid physical processes through parameterizations (or closure) in numerical simulations with quantified uncertainty are critical for resolving the coarse-grained partial differential equations that govern many problems ranging from weather and climate prediction to turbulence simulations. Recent advances have seen machine learning (ML) increasingly applied to model these subgrid processes, resulting in the development of hybrid physics-ML models through the integration with numerical solvers. In this work, we introduce a novel framework for the joint estimation of physical parameters and machine learning parameterizations with uncertainty quantification. Our framework incorporates online training and efficient Bayesian inference within a high-dimensional parameter space, facilitated by differentiable programming. This proof of concept underscores the substantial potential of differentiable programming in synergistically combining machine learning with differential equations, thereby enhancing the capabilities of hybrid physics-ML modeling.

## 1 Introduction

Weather and climate models are critical tools for predicting weather patterns, understanding climate change, and informing future environmental policies and strategies IPCC (2021). Central to these models is the challenging task of precisely solving time-dependent parametric partial differential equations (PDEs) that encapsulate the intricate dynamics of Earth systems. A key challenge in these models stems from the chaotic and multi-scale nature of atmospheric and oceanic processes. Owing to computational constraints, these models are typically simulated on coarse meshes ($O(10)$km in the horizontal), leading to miss-representation of crucial sub-grid scale processes (Schneider et al., 2017). Yet, the mere increase in resolution is insufficient, as the set of PDEs remains unclosed due to the absence of governing equations for certain critical, yet poorly understood or unknown processes, such as the carbon cycle (Trugman et al., 2018) or microphysics. These gaps introduce significant challenges to weather and climate projection Nathaniel et al. (2024); Bony et al. (2015), and underscore the need for a robust methodology to capture and couple all these dynamics that are not directly resolved or described, which is called closure or parameterization (Randall et al., 2003). Most traditional parameterization schemes contain an empirical functional relationship with tunable physical parameters (Smagorinsky, 1963; Siebesma et al., 2007). These parameterization schemes contribute to model uncertainty (Draper, 1995). Estimating the physical parameters of interest, regarded as an inverse problem, can be approached by variational data assimilation (Smith et al., 2009) ensemble methods such as Kalman filter ensemble methods (Evensen, 2009; Cleary et al., 2021) and Monte-Carlo based approaches (Yang et al., 2020), but the quality of inferred models is challenged by the dynamical systems' strong nonlinearity (Cheng et al., 2023), and heuristic assumptions behind traditional parameterization schemes (Gentine et al., 2018). The latter limitation has spurred the use of machine learning for modeling sub-grid scale dynamics from high-resolution simulations to emulate the coarse one (Rasp et al., 2018; Bhouri et al., 2023; Zanna & Bolton, 2020). Kalman

based methods and variational methods can presumably be used to infer physical and machine learning parameters based on uncertain observations (Evensen, 2009). However, the strong non-linearity of the underlying models presents a challenge to the Gaussian and near-linear assumption underlying Kalman filtering (Van Leeuwen et al., 2015) and the dimensionality of the problem (with millions of degrees of freedom for neural network), limits their applicability and performance. On the other hand, differentiable modeling has been showing great promise in integrating machine learning and physical models (Shen et al., 2023). It allows taking the derivatives within numerical errors to any model parameters whether neural network based on physically based, thus permitting the use of modern optimization techniques (backpropagation) for model inference. For parameterization, online training of neural networks (NN) with differentiable solver through target trajectories offers numerous benefits. These include enhanced numerical stability and accuracy (Frezat et al., 2022; Qu & Shi, 2023), flexibility to integrate variational data assimilation (Farchi et al., 2023; Qu & Shi, 2023) without Gaussian assumption, and efficient uncertainty quantification facilitated by gradients of the numerical solver (Yang et al., 2020; Bhouri & Gentine, 2022). The recent development of differentiable general circulation models (NeuralGCM, Kochkov et al. (2023)) signals a promising future for scaling up from surrogate models like the Lorenz systems to larger-scale, more realistic systems.

In this work, we consider a hybrid model that contains poorly known physical parameter values, and a neural network for sub-grid scale parameterization of turbulence. We approach the joint estimation of physical parameters and machine learning parameters with quantified uncertainty, framed as a Bayesian inverse problem, through a 2-stage approach enabled by differentiable programming. An initial estimate of the set of parameters is obtained using stochastic gradient-based optimization on temporally sparse trajectories. Then, we perform Bayesian inference of the set of parameters using stochastic gradient Hamiltonian Monte Carlo (SG-HMC) (Chen et al., 2014), also through the set of temporally sparse trajectories. As a proof of concept, the proposed approach is applied to a two-layer quasi-geostrophic model to illustrate the potential of next generation Earth System Models combining Bayesian ML and physics using a differentiable programming framework.

## 2 APPROACH

An abstraction of the coarse-grained dynamical system for climate and weather prediction can be represented by the following differential equation:

$$\frac{d\overline{\mathbf{X}}}{dt} = F(\overline{\mathbf{X}}; \boldsymbol{\theta}_{phy}) + G(\overline{\mathbf{X}}; \boldsymbol{\theta_1}), \tag{1}$$

with appropriate initial and boundary conditions. Here, $\overline{\mathbf{X}}$ denotes the estimate of true physical states $\mathbf{X}$. The function $F$ encapsulates the resolved dynamics depending on physical parameters $\boldsymbol{\theta}_{phy} \in \mathbb{R}^{d_1}$. The function $G$ models unknown or sub-grid scale dynamics as a function of $\overline{\mathbf{X}}$ and parameters $\boldsymbol{\theta_1}$. In this study, we model $G$ as a neural network with parameter $\boldsymbol{\theta}_{NN} \in \mathbb{R}^{d_2}$. Typically, $d_2$ is much larger than $d_1$, reflecting the higher dimensionality of parameter space in neural networks compared to physical parameters and traditional parameterization schemes. Given numerical solver time step $\Delta t$ and initial value $\mathbf{X}_{t_0}$, $n$ step integration using an explicit numerical scheme results in $\overline{\mathbf{X}}_{t_0+n\Delta t} = \mathcal{M}^n(\mathbf{X}_{t_0}; \boldsymbol{\theta}_{phy}, \boldsymbol{\theta}_{NN})$, where $\mathcal{M}$ is a differentiable numerical solver utilized to evolve Equation 1, and $\mathcal{M}^n$ denotes the n-fold composition of M with itself. Assuming one ground truth observation is available every $\Delta T = k\Delta t$ (i.e. every $k$ model time steps), a temporally sparse trajectory of $N+1$ ground truth data points is represented as $\{\mathbf{X}_{t_0+i\Delta T}\}_{i=0}^N$, and the corresponding forecast trajectory obtained from solving Equation 1 is denoted as $\{\overline{\mathbf{X}}_{t_0+i\Delta T}(\boldsymbol{\theta}_{phy}, \boldsymbol{\theta}_{NN})\}_{i=0}^N = \{\mathcal{M}^{ik}(\mathbf{X}_{t_0}; \boldsymbol{\theta}_{phy}, \boldsymbol{\theta}_{NN})\}_{i=0}^N$.

**Online deterministic training:** An estimate of $\{\boldsymbol{\theta}_{phy}^*, \boldsymbol{\theta}_{NN}^*\}$, used subsequently to initialize the Markov Chain, is obtained by mini-batch gradient-based optimization of a loss function,

$$\mathcal{J}(\boldsymbol{\theta}_{phy}, \boldsymbol{\theta}_{NN}) = \frac{1}{|I|} \sum_{t_0 \in I} \mathcal{L}\left(\{\mathbf{X}_{t_0+i\Delta T}\}_{i=0}^N, \{\overline{\mathbf{X}}_{t_0+i\Delta T}(\boldsymbol{\theta}_{phy}, \boldsymbol{\theta}_{NN})\}_{i=0}^N\right), \tag{2}$$

where $I$ is a random batch of ground truth trajectories' initial time-steps from training dataset $\mathcal{D}$, and $\mathcal{L}(\cdot, \cdot)$ evaluates the distance between a ground truth trajectory and the corresponding forecast. In

each training iteration, parameters $\boldsymbol{\theta}_{phy}$ and $\boldsymbol{\theta}_{NN}$ are updated based on $\partial \mathcal{J}/\partial \boldsymbol{\theta}_{phy}$ and $\partial \mathcal{J}/\partial \boldsymbol{\theta}_{NN}$, respectively. This separation permits the potential deployment of distinct learning rates and stochastic gradient-based algorithms for each parameter type. Moreover, one may choose to cease the update of $\boldsymbol{\theta}_{phy}$ upon convergence, focusing solely on fine-tuning $\boldsymbol{\theta}_{NN}$, especially considering the high dimensionality of the latter. The gradients can be obtained conveniently when $\mathcal{M}$ is written in programming frameworks that support automatic differentiation, such as JAX (Bradbury et al., 2018), PyTorch (Paszke et al., 2019) and Julia (Bezanson et al., 2017).

**Bayesian inference and uncertainty propagation:** In contrast to traditional Markov-Chain Monte Carlo (MCMC) sampling methods, which are computationally intensive for large-scale Bayesian inference (Van Ravenzwaaij et al., 2018), Hamiltonian Monte Carlo (HMC) methods offer an efficient means to sample high-dimensional parameter spaces (Neal et al., 2011). To circumvent directly computing the costly gradient of the potential energy over the whole dataset, we adopt the stochastic gradient HMC (SG-HMC) Chen et al. (2014), which approximates the gradient by evaluating the likelihood on mini-batches. A Bayesian hierarchical approach is applied to quantify the uncertainty. We combine $\{\boldsymbol{\theta}_{phy}, \boldsymbol{\theta}_{NN}\}$ as a set of uncertain model parameters $\boldsymbol{\theta} \in \mathbb{R}^{d_1+d_2}$ with a prior parameterized by $\lambda$ that encodes our prior knowledge about $\boldsymbol{\theta}$. Another random variable $\gamma$ is introduced to quantify the quality of data. The likelihood is constructed as follows:

$$p\left(\{\mathbf{X}_{t_0+i\Delta T}\}_{i=1}^N \mid \boldsymbol{\theta}, \gamma\right) = \prod_{t_0 \in I} \mathcal{N}\left(\{\mathbf{X}_{t_0+i\Delta T}\}_{i=1}^N \mid \{\mathcal{M}^{ik}(\mathbf{X}_{t_0}; \boldsymbol{\theta})\}_{i=1}^N, \gamma^{-1}\right), \tag{3}$$

where $I$ denotes a random batch of initial times of ground truth trajectories from training dataset $\mathcal{D}$, and $\mathcal{N}$ represents the probability density function for normal distribution. The posterior distribution is then formulated as:

$$p\left(\boldsymbol{\theta}, \gamma, \lambda \mid \{\mathbf{X}_{t_0+i\Delta T}\}_{i=0}^N\right) \propto p\left(\{\mathbf{X}_{t_0+i\Delta T}\}_{i=0}^N \mid \boldsymbol{\theta}, \gamma\right) p(\boldsymbol{\theta} \mid \lambda) p(\lambda) p(\gamma), \tag{4}$$

with specifics on the selections of priors detailed in Appendix B. Importantly, the posterior distribution does not depend on initial time $t_0$ of trajectories as long as the inference is conducted in statistically quasi steady state of the system, taking into account the ergodicity. The likelihood formulation Equation 3 emphasizes the need for a differentiable PDE solver $\mathcal{M}$ as SG-HMC necessitates the computation of the gradient of the log-likelihood with respect to $\boldsymbol{\theta}$. The Markov Chain sampling is initialized with $\boldsymbol{\theta}^* = \{\boldsymbol{\theta}_{phy}^*, \boldsymbol{\theta}_{NN}^*\}$ to favor a short transient phase and improve sampling robustness. The predictive posterior distribution of a forecast at time $t$, denoted by $\mathbf{X}^*(t)$, is then given by

$$p(\mathbf{X}^*(t) \mid \mathcal{D}, \mathbf{X}_{t_0}, t) = \int p(\mathbf{X}^*(t) \mid \boldsymbol{\theta}, \gamma, \lambda, \mathbf{X}_{t_0}, t) p(\boldsymbol{\theta}, \gamma, \lambda \mid \mathcal{D}) d\boldsymbol{\theta} d\gamma d\lambda, \tag{5}$$

allowing us to sample $\mathbf{X}^*(t)$ for a given initial state $\mathbf{X}_{t_0}$, assuming $t - t_0 = l\Delta t$ for some integer $l$, as depicted in the following equation:

$$\mathbf{X}^*(t) = \mathcal{M}^l(\mathbf{X}_{t_0}; \boldsymbol{\theta}) + \epsilon, \epsilon \sim \mathcal{N}(0, \gamma^{-1}), \{\boldsymbol{\theta}, \gamma, \lambda\} \sim p(\boldsymbol{\theta}, \gamma, \lambda | \mathcal{D}). \tag{6}$$

The posterior mean and variance of $\mathbf{X}^*(t)$ can be approximated from SG-HMC samples as detailed in Appendix B.

## 3 EXPERIMENTS AND RESULTS

The proposed framework is applied to the two-layer quasi-geostrophic equations with rigid lid approximation and flat bottom topography, as implemented in PyQG JAX port, which supports automatic differentiation (Otness et al., 2023). Detailed descriptions of the model's governing equation, the coarse-grained equation, sub-grid scale terms targeted for parameterization, and parameter specifics are detailed in Appendix A. "Ground truth" data is generated from simulations on a $256 \times 256$ mesh covering a $1,000 \text{km} \times 1,000 \text{km}$ domain, with convergence testing outlined in Ross et al. (2023). Simulations span 10 years at a timestep of $\Delta t = 1 \text{hour}$, using third-order Adams-Bashforth time stepping, and the initial 5 years are excluded as spin-up. 60 simulations are used for deterministic training and Bayesian inference, with additional 10 for testing. Finally, the data is coarse-grained to a $32 \times 32$ mesh using a sharp spectral cutoff filter (details in Appendix A).

The physical parameter set, $\boldsymbol{\theta}_{phy} = \{\delta, U_1\}$, includes the layer thickness ratio $\delta$ and the upper layer background velocity $U_1$, with true values of $\{0.25, 0.025\}$ and initial guess of $\{0.01, 0.001\}$. Sub-grid total tendency was modeled using a convolutional neural network (CNN) that takes the upper and bottom layer vorticities as inputs, with an architecture detailed in Appendix C. Temporally sparse trajectories of just 20 daily observations at one observation per day ($\Delta T = 24\Delta t$), were used for online deterministic training. The mean squared error (MSE) served as the loss function. Online deterministic training proceeds with updating both $\boldsymbol{\theta}_{phy}$ and $\boldsymbol{\theta}_{NN}$ until convergence of $\boldsymbol{\theta}_{phy}$, followed by refinement of $\boldsymbol{\theta}_{NN}$ with fixed $\boldsymbol{\theta}_{phy}$. The dimensional structure of the samples generated via SG-HMC mirrors that utilized during the deterministic training phase. Details of online deterministic training and SG-HMC sampling, such as learning rates, optimizer selection and HMC step size, are provided in Appendix C. The training and validation curves and history of $\boldsymbol{\theta}_{phy}$ are shown in Figure 2 in Appendix D. The inferred physical parameters from the deterministic training, $\boldsymbol{\theta}_{phy}^* = \{0.25636, 0.02535\}$, align closely with their true values, exhibiting relative errors of $\{2.54\%, 1.43\%\}$.

To assess the efficacy of the inferred physical parameters and neural network parameterizations, equation 1 is solved using a ground truth initial condition from the statistically steady state (i.e., post-year 5), over a one-year period (8,640 $\Delta t$). Evaluation metrics include the coefficient of determination ($R^2$), Mean Squared Error (MSE), and total kinetic energy. For a comprehensive analysis of long-term behavior, the empirical distribution of the upper layer potential vorticity was examined over the prediction period's final 100 days. Performance comparisons were drawn between our framework's deterministic, maximum *a posteriori*(MAP) and posterior mean estimates, against traditional Smagorinsky schemes and scenarios without parameterization. The uncertainty was quantified through a 2 posterior standard deviation band, encapsulating 95% of variability, as depicted in Figure 1 . For the prediction window up to 4,000 hours, our framework's deterministic esti-

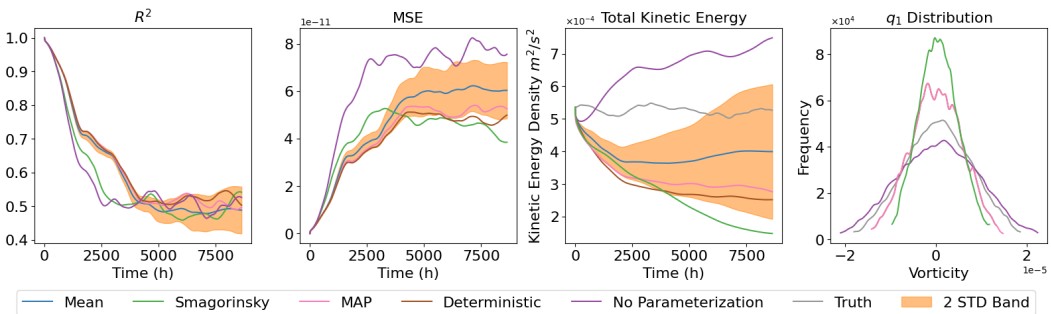

Figure 1: Metrics evaluating the performance of online predictions over 1 year period.

mate outperform both the no-parameterization approach and the Smagorinsky scheme in terms of achieving the highest $R^2$ and lowest MSE. The MAP estimate's performance closely mirror that of the deterministic estimate. Within the initial 2,500 hours, the posterior mean showcase commendable accuracy with a minimal spread of uncertainty. The Smagorinsky parameterization exhibited over-dissipation, as shown by the continuous decline in total kinetic energy and alterations in the long-term vorticity distribution. Conversely, simulations at low resolution displayed unphysical increases in total kinetic energy, hinting at potential numerical instability for extended simulations. In contrast, our framework's deterministic, MAP and posterior mean predictions manage to better conserve total kinetic energy, albeit with a significant uncertainty spread. These outcomes underscore the viability of the proposed approach in enhancing weather and climate model predictions through efficient parameterization and uncertainty quantification. Additional results are available in Appendix D.

## 4 CONCLUSION AND DISCUSSION

Our proposed framework demonstrates the potential of integrating Bayesian differentiable programming with physical parameter inference and machine learning parameterization. This integration, complemented by efficient Bayesian inference, paves the way for more accurate and reliable scien-

tific simulations and knowledge discovery. Current efforts are directed towards optimizing online training strategies, specifically exploring the optimal selection and configuration of ground truth trajectories, including their length and temporal sparsity. Such design considerations are closely tied to the characteristics of the system under study, aiming to balance the reduction of temporal correlation with the challenges of gradient backpropagation over extensive, sparse trajectories. Given the potential inaccuracies in gradient estimates from SG-HMC in large datasets, especially where high-precision scientific simulation is required, alternatives such as Control Variate Gradient HMC (CVG-HMC) (Zou & Gu, 2021) can be considered for gradient estimation improvements. Our future work will extend to accommodate spatially sparse ground truth data and noise, and integrate state inference, as considered in Qu et al. (2024). For various systems, inferring parameters and parameterizations simultaneously may exhibit the equifinality issue, highlighting the importance of continuous effort to further improve the proposed framework's generalizability. Through these endeavors, we aim to refine and expand the capabilities of our methodology, contributing to the advancement of artificial intelligence for scientific modeling.

## ACKNOWLEDGMENTS

We acknowledge funding from NSF through the Learning the Earth with Artificial intelligence and Physics (LEAP) Science and Technology Center (STC) (Award #2019625).We would also like to acknowledge high-performance computing support from Derecho (doi:10.5065/qx9a-pg09) provided by NCAR's Computational and Information Systems Laboratory, sponsored by NSF.

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

## A   QG MODEL PARAMETERS AND DATA GENERATION

We consider the two-layer quasi-geostrophic(QG) equation illustrating flows driven by baroclinic instability of a background velocity shear $U_1$ - $U_2$ with rigid lid approximation and flat bottom topography:

$$
\begin{aligned}
\frac{\partial q_1}{\partial t} + J(\psi_1, q_1) + \beta \frac{\partial \psi_1}{\partial x} + U_1 \frac{\partial q_1}{\partial x} &= 0, \\
\frac{\partial q_2}{\partial t} + J(\psi_2, q_2) + \beta \frac{\partial \psi_2}{\partial x} + U_2 \frac{\partial q_2}{\partial x} &= -r_{ek} \nabla^2 \psi_2,
\end{aligned}
\tag{7}
$$

where $J(\cdot, \cdot)$ is the horizontal Jacobian, $q_i$, $\psi_i$ is the layer-$i$ potential vorticity and stream function, respectively. They are related through

$$
\begin{aligned}
q_1 &= \nabla^2 \psi_1 + \frac{1}{(r_d)^2(1+\delta)}(\psi_2 - \psi_1), \\
q_2 &= \nabla^2 \psi_2 + \frac{\delta}{(r_d)^2(1+\delta)}(\psi_1 - \psi_2).
\end{aligned}
\tag{8}
$$

The physical meaning of physical parameters and their setting for generating ground truth can be found in Table 1. The parameters to be inferred is a subset of the physical parameters, denoted by

Table 1: List of physical parameters

| PARAMETER | DESCRIPTION | VALUE |
|---|---|---|
| $\beta$ | Rossby Parameter | $1.5 \times 10^{-11}$ |
| $r_{ek}$ | Linear Botton Drag Coefficient | $5.787 \times 10^{-7}$ |
| $r_d$ | Deformation Wavenumber | $1.5 \times 10^{4}$ |
| $U_1$ | Upper Layer Background x-axis velocity | $2.5 \times 10^{-2}$ |
| $U_2$ | Lower Layer Background x-axis valocity | $0$ |
| $\delta$ | Layer Thickness Ratio $H_1/H_2$ | $2.5 \times 10^{-1}$ |

$\theta_{phy}$. Following Ross et al. (2023), we approximate the effect of grid by a coarse-graining filter $\overline{(\cdot)}$, and the sub-grid dynamics to be parameterized is the sub-grid total tendency

$$S_i = \overline{\frac{\partial q_i}{\partial t}} - \frac{\partial \overline{q}_i}{\partial t}, i = 1, 2, \tag{9}$$

The filter applied is a sharp spectral truncation filter as following:

$$\hat{\overline{q}}_\kappa = \begin{cases} \hat{q}_\kappa, & \kappa < \kappa^c, \\ \hat{q}_\kappa \cdot e^{-23.6(\kappa - \kappa_c)^4 \Delta x_{\text{LowRes}}^4}, & \kappa \geq \kappa^c, \end{cases} \tag{10}$$

where $\hat{q}_\kappa$ is the Fourier transformation of vorticity at wave number $\kappa$, $\kappa_c$ is the cutoff threshold and $\Delta x_{\text{LowRes}}$ is the spatial resolution of low-resolution simulations. The sub-grid totalt tendency is not available in low-resolution simulations since the first term at the right-hand-side of Equation 9 is a filtered ground truth/high-resolution total tendency. Therefore, we use a neural network with parameter $\boldsymbol{\theta}_{NN}$ to model it as a function of low-resolution variable.

## B  PRIORS AND POSTERIOR STATISTICS

Following Bhouri & Gentine (2022), the choices of priors are given by

$$\boldsymbol{\theta} \mid \lambda \sim \text{Laplace}(\boldsymbol{\theta} \mid 0, \lambda^{-1}), \tag{11}$$

$$\log \lambda \sim \text{Gamma}(\log \lambda \mid \alpha_1, \beta_1), \tag{12}$$

$$\log \gamma \sim \text{Gamma}(\log \gamma \mid \alpha_2, \beta_2), \tag{13}$$

where $\alpha_1, \alpha_2, \beta_1, \beta_2$ are hyperparameters. The the use of logarithm transformation is to ensure $\lambda$ and $\gamma$ are always positive. The posterior mean and variance are:

$$\mu_{\mathbf{X}^*}(t) = \int \mathcal{M}^l(\mathbf{X}_{t_0}; \boldsymbol{\theta}) p(\boldsymbol{\theta} \mid \mathcal{D}) d\boldsymbol{\theta} \approx \frac{1}{N_s} \sum_{i=1}^{N_s} \mathcal{M}^l(\mathbf{X}_{t_0}; \boldsymbol{\theta}_i), \tag{14}$$

$$\sigma_{\mathbf{X}^*}^2(t) = \int \left(\mathcal{M}^l(\mathbf{X}_{t_0}; \boldsymbol{\theta}) - \mu_{\mathbf{X}^*}(t)\right)^2 p(\boldsymbol{\theta} \mid \mathcal{D}) d\boldsymbol{\theta} \approx \frac{1}{N_s} \sum_{i=1}^{N_s} \left(\mathcal{M}^l(\mathbf{X}_{t_0}; \boldsymbol{\theta}_i) - \mu_{\mathbf{X}^*}(t)\right)^2 \tag{15}$$

Here $N_s$ is the number of samples that approximates the posterior distributions obtained through SG-HMC sampling.

## C  ONLINE DETERMINISTIC TRAINING AND BAYESIAN INFERENCE DETAILS

The sub-grid total tendency was parameterized using a convolutional neural network (CNN), detailed in Table 2. This CNN incorporates periodic padding in each layer, includes bias terms, omits batch normalization, and employs the ReLU activation function. The parameter space of the CNN is dimensioned at $d_2 = 113,766$. Implementation was carried out using the Flax library (Heek et al., 2023).

Table 2: Neural Network Architecture

| Convolution Layer Number | Output Channels | Kernel Size |
|---|---|---|
| 1 | 128 | (3,3) |
| 2 | 64 | (3,3) |
| 3 | 32 | (3,3) |
| 4 | 32 | (3,3) |
| 5 | 32 | (3,3) |
| 6 | 2 | (3,3) |

In the online deterministic training phase, we employed two separate AdaBelief optimizers (Zhuang et al., 2020) for optimizing $\theta_{NN}$ and $\theta_{phy}$, each with its distinct learning rate schedule. Specifically, $\theta_{phy}$ utilized an exponential decaying learning rate, starting at 0.01 and decreasing to 0.001 with a decay rate of 0.9. Similarly, for $\theta_{NN}$, an exponential decaying learning rate was applied, commencing at 0.0005 and diminishing to 0.0001 with a decay rate of 0.95. Convergence of $\theta_{phy}$ was observed near the 50-epoch mark, at which point it was held constant to exclusively continue the training of the CNN for an additional 50 epochs.

During the SG-HMC sampling phase, the hyperparameters $\alpha_1$, $\beta_1$, $\alpha_2$, and $\beta_2$ within the prior distributions are all assigned a value of 1. The leapfrog step size for SG-HMC, denoted as $\epsilon_{HMC}$, is set to $5 \times 10^{-5}$, with the number of leapfrog steps per iteration fixed at $L = 10$. The sampling process is conducted over 2,000 iterations.

## D ADDITIONAL RESULTS

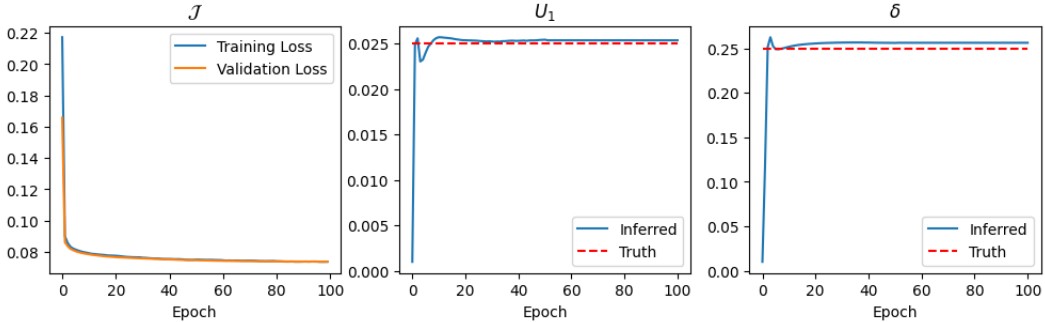

Figure 2: Curves for training loss, validation loss, $U_1$ and $\delta$. Note that the values are averaged over all the batches within an epoch. After 50 epoches, $U_1$ and $\delta$ are fixed.

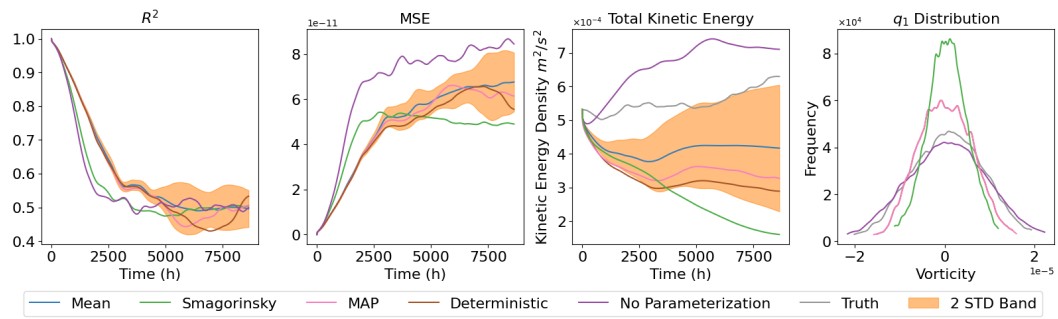

Figure 3: Metrics evaluating the performance of online predictions over 1 year period. Same as Figure 2, but evaluated on another test case.

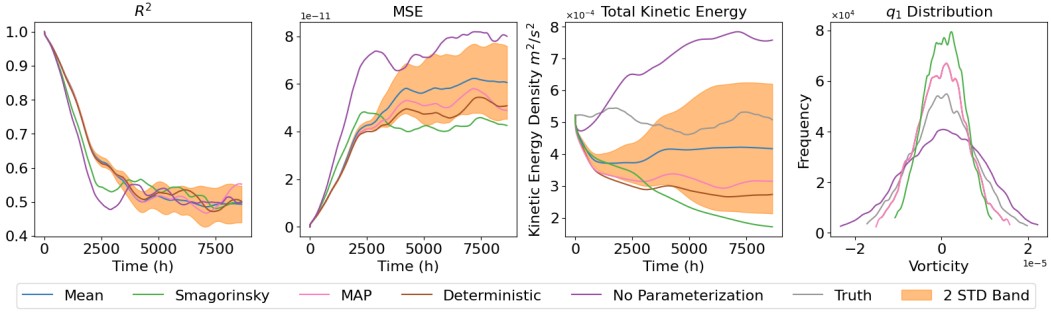

Figure 4: Metrics evaluating the performance of online predictions over 1 year period. Same as Figure 2 and 3, but evaluated on another test case.

