# OpenReview forum: "Joint Parameter and Parameterization Inference with Uncertainty Quantification Through Differentiable Programming"
_ICLR.cc/2024/Workshop/AI4DiffEqtnsInSci — AI4DiffEqtnsInSci @ ICLR 2024 Poster_

### Official Review · Reviewer_GJgg · 2024-02-23
**Parameter inference for weather and climate models.**

**Rating:** 7
**Confidence:** 3

**Review:**

Summary: This paper proposed a novel method for parameter estimation and uncertainty quantification which is then applied to meaningful problems in weather and climate modeling.

Strengths and Weaknesses:

* Should add (ours) to the legend in Figure 1 to clearly indicate the author's proposed methods.

[+] The paper is well-motivated and clearly presented. It provides a clear description of the problem and the proposed solution. It is very well written, with comprehensive references to relevant material.

[+] The uncertainty quantification is a nice benefit over the baseline approaches (Smagorinsky & no-parameterization schemes)

[-] It would be nice to have an overview figure or algorithm of the proposed framework to show the flow from data to initialization to model to prediction and what each step entails.

[-] It would be nice to have a ground truth solution plot, either in the appendix or main text, so readers can better visualize the problem being solved. From my understanding, the author used the problem in Ross et. al. (2023). In that paper, Figure 1 gives a nice visual of the vorticity. This would help connect readers back to the application of the author's method to Earth's weather and climate predictions.

Conclusion: The paper is concise and to the point. While there could be some improvement in the presentation of the problem and methodology, the discussion is still clear. I recommend accepting the paper and believe it is important to develop better methodologies for this class of problems after reading the paper. While the proposed method didn't excel in all metrics over all times, it shows that it is still a complex and difficult problem to solve, and this work makes a nice step in the right direction.

---

### Official Review · Reviewer_KThv · 2024-02-25
**Advancing hybrid modeling parameterizations with uncertainty estimates**

**Rating:** 8
**Confidence:** 4

**Review:**

Summary
------------------
This paper proposes a hybrid modeling approach that allows learning parameters and parametrizations in Earth system models, with the additional advantage of delivering uncertainty estimates. The approach is showcased in a toy example but seems promising and advancing in the right direction.  Authors consider a hybrid model that contains poorly known physical parameters and a neural network for sub-grid scale parameterization. They approach the joint estimation of physical and ML model parameters with quantified uncertainty, framed as a Bayesian inverse problem. The work, even if in its early stages, shows promise, and if thoroughly advanced, it could have a strong impact on the Earth and climate science community. It goes without saying that the approach, in a similar vein to NeuralGCM, would not only learn parametrizations but scale up from surrogate models like the Lorenz systems to larger-scale, more realistic scenarios.

Evaluation
------------------

**Quality:**
The paper demonstrates a robust research methodology, technical depth, and experimental rigor. It addresses a significant challenge in weather and climate modeling and proposes a novel framework for joint estimation and uncertainty quantification of physical parameters and machine learning parameterizations. The experiments are well-designed, using appropriate simulations and evaluation metrics, and the results are thoroughly analyzed. Overall, a high-quality piece of work.

**Clarity:**
The paper effectively communicates complex concepts in a clear and understandable manner. The introduction provides necessary background information, and the methodology section is logically organized with illustrative equations and figures. The experimental setup and results are presented in a structured format, making it easy to follow the research workflow and understand the findings.

**Originality:**
The paper demonstrates originality by proposing a novel framework for addressing a pressing challenge in weather and climate modeling. Integrating differentiable programming with Bayesian inference represents a unique approach to enhancing hybrid physics-ML modeling, with potential implications for advancing key problems in Earth system science.

**Significance:**
The paper's findings are highly significant for weather and climate modeling, offering potential improvements in prediction accuracy and reliability. The experimental results demonstrate promising performance over traditional parameterization schemes, highlighting practical implications for environmental science and artificial intelligence in scientific modeling.

Clarifications
------------------
- clarify what you mean by "The superscript in M n denotes function composition"

- some comments on computational complexity, a bit of ablation study on all parts (integration, HMC, model runs), and some words about future schemes

- comments about calibration of uncertainties would be welcome.

- what are "Smagorinsky schemes"?

- unlike variational approaches, MC schemes have shown great advantages in dealing with multimodal distributions but are more costly (see Svendsen 2023, https://doi.org/10.1007/s10994-021-05999-4). some discussion about this in the context of learning dynamics?

- when learning params and NN params, which are unevenly distributed and dimensional, some words of caution should be expected, both because I presume the optimization may become quite unstable and also because of the risk of running into identifiability problems, right?

- Why MSE, especially in such a dimensional uneven setting?

Typos and grammar issues
------------------

- Fix notation around (1); \theta_1 --> \theta_NN (also take the opportunity to harmonize dimensionality notation, e.g. d_phys and d_nn, and eventually give rough numbers of those beyond the d_nn>>d_phys for scientists that are not in the field)

- several typos: "Hamiltonian Monte Calro (HMC)", "which approximate_S_ the gradient", "Peter Jan Van Leeuwen," double author?, "maximum a posteriori_(MAP)", etc. Please give a grammar pass to the text and clean your bib (this is a nice tool, btw: https://flamingtempura.github.io/bibtex-tidy/).

---

### Official Review · Reviewer_it2S · 2024-02-25
**Joint probabilistic inference in hybrid partial differential equations for climate modeling: A promising proof of concept for scalable Bayesian inference with differentiable programming.**

**Rating:** 8
**Confidence:** 5

**Review:**

Summary:
The authors propose and explore an approach to do joint inference over a hybrid (semi-parametric) partial differential equation. In a two-step approach, they first fit both the non-parametric and the parametric part jointly with gradient descent and use the result as a starting point for SG-HMC. One key aspect of the pipeline for the integration with numerical solvers is differentiable programming.

Pros:
- The approach is set up for realistic problems of dynamical systems for climate and weather predictions and as such highly relevant.
- The manuscript is well written and the authors do a great job in clearly explaining the different layers of their approach.
- The work shows a proof of concept highlighting the advantage of the approach over traditional and non-parametric methods.
- A discussion on ways forward to improve it or potential issues within semi-parametric modeling, such as equifinality/non-identifiability, are briefly mentioned.

Cons:
- A more thorough explanation of the results for the concrete problem and their consequences would be desirable (though I understand that it goes beyond what fits into a workshop paper of 4 pages).

Open questions:
- It is interesting to me how this approach would compare to other scalable approaches of deep uncertainty quantification, such as deep ensembles or variational inference, both in performance and posterior distributions.
- As we are doing joint inference over the parameters and the non-parametric part, it would also be nice to show the posterior distributions over the $\delta$ and $U_1$ parameters.
- Given that the data is synthetic, is there a specific reason why the deterministic estimator does not converge to the exact $\delta$ and $U_1$ value? Does it average out with multiple fitting runs, or is there a structural positive bias? Does it persist in the posterior distribution?
- What is the no-parameterization approach?
- The results show a clear advantage of the approach over traditional and non-parametric approaches up to roughly Time 3500h. After that, the Smagorinsky parametrization seems to take the lead. How can this be explained, and is there a need and/or a way to improve the long-term performance?

---

### Meta-Review · Area_Chair_JM3a · 2024-03-01

**Recommendation:** Accept (Poster)

**Metareview:**

All reviewers unanimously agree on acceptance. However, author should address the concerns on the camera-ready version

---

### Decision · Program_Chairs · 2024-03-01

Accept (Poster)